# DEEP ENSEMBLES FOR GRAPHS WITH HIGHER-ORDER DEPENDENCIES

**Steven J. Krieg, William C. Burgis, Patrick M. Soga, & Nitesh V. Chawla***
Lucy Family Institute for Data and Society
University of Notre Dame
Notre Dame, IN 46556
{skrieg,wburgis,psoga,nchawla}@nd.edu

## ABSTRACT

Graph neural networks (GNNs) continue to achieve state-of-the-art performance on many graph learning tasks, but rely on the assumption that a given graph is a sufficient approximation of the true neighborhood structure. When a system contains higher-order sequential dependencies, we show that the tendency of traditional graph representations to underfit each node's neighborhood causes existing GNNs to generalize poorly. To address this, we propose a novel **Deep Graph Ensemble** (DGE), which captures neighborhood variance by training an ensemble of GNNs on different neighborhood subspaces of the same node within a higher-order network representation. We show that DGE consistently outperforms existing GNNs on semisupervised and supervised tasks on six real-world data sets with known higher-order dependencies, even under a similar parameter budget. We demonstrate that diverse and accurate base classifiers are central to DGE's success, and discuss the implications of these findings for future work on ensembles of GNNs.

## 1 INTRODUCTION

Graph neural networks (GNNs) solve learning tasks by propagating information through each node's neighborhood in a graph (Zhou et al., 2020; Wu et al., 2020). Most present work on GNNs assumes that a given graph is a sufficient approximation of the underlying neighborhood structure. But a growing body of work has challenged this assumption by showing that traditional graphs often cannot capture the higher-order structure and dynamics that govern many real-world systems (Lambiotte et al., 2019; Battiston et al., 2020; Porter, 2020; Torres et al., 2021; Battiston et al., 2021). In the present work, we couple GNNs with a specific family of graphs, **higher-order networks** (HONs), which encode sequential **higher-order dependencies** (i.e., conditional probabilities that cannot be explained by a first-order Markov model) in a graph structure. A traditional graph, which we call a **first-order network** (FON), represents a system by decomposing it into a set of pairwise edges, so the only way to infer polyadic interactions is via transitive paths over adjacent nodes. When higher-order dependencies are present, these Markovian paths underfit the true neighborhood (Scholtes, 2017) and can thus produce many false positive interactions between nodes (Lambiotte et al., 2019). To address this limitation, Xu et al. (2016) proposed a HON that creates conditional nodes to more accurately encode the observed higher-order interactions. By preserving this additional information in the graph structure, HONs have produced new insights in studies of user behavior (Chierichetti et al., 2012), citation networks (Rosvall et al., 2014), human mobility and navigation patterns (Scholtes et al., 2014; Peixoto & Rosvall, 2017), the spread of invasive species Saebi et al. (2020b), anomaly detection (Saebi et al., 2020d), disease progression (Krieg et al., 2020b), and more (Koher et al., 2016; Peixoto & Rosvall, 2017; Scholtes, 2017; Lambiotte et al., 2019; Saebi et al., 2020a). However, their use with GNNs has not been thoroughly explored.

As Figure 1 illustrates, the tendency of FONs to underfit has consequences for GNNs, which typically compute representations by recursively pooling features from each node's neighbors. In order

---

*Corresponding author.

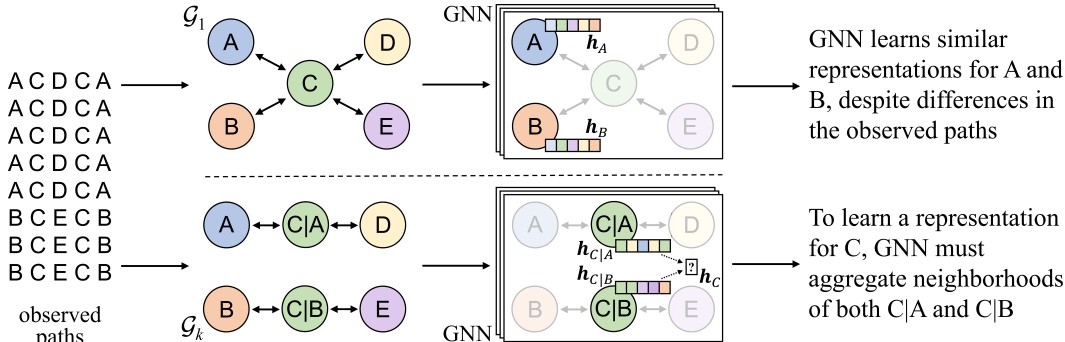

Figure 1: A toy example of challenges faced by GNNs in modeling systems with higher-order dependencies. A FON ($\mathcal{G}_1$) underfits the higher-order dependencies in the observed paths. Consequently, a GNN will learn similar representations for A and B, since they share the same 2-hop neighborhood in $\mathcal{G}_1$. A HON ($\mathcal{G}_k$, with $k = 2$ in this example) uses conditional nodes to encode higher-order dependencies. For example, node C|A represents the observed dependency that C only interacts with D when it also interacts with A (note that in real-world systems, $\mathcal{G}_k$ rarely breaks the graph into multiple components). However, computing a representation for C then requires a GNN to aggregate multiple local neighborhoods. Colors depict node features.

to maximize GNN performance, we must ensure that local neighborhoods capture the true distribution of interactions in the system. To enable GNNs to utilize the additional information encoded in HONs, we propose a novel **Deep Graph Ensemble** (DGE), which uses independent GNNs to exploit variance in higher-order node neighborhoods and learn effective representations in graphs with higher-order dependencies. The **key contributions** of our work include:

1. We analyze the data-level challenges that fundamentally limit the ability of existing GNNs to learn effective models of systems with higher-order dependencies.
2. We introduce the notion of neighborhood subspaces by showing that neighborhoods in a HON are analogous to feature subspaces of first-order neighborhoods. Borrowing from ensemble methods, we then propose DGE to exploit the variance in these subspaces.
3. We experimentally evaluate DGE against eight state-of-the-art baselines on six real-world data sets with known higher-order dependencies, and show that, even with similar parameter budgets, DGE consistently outperforms baselines on semisupervised (node classification) and supervised (link prediction) tasks.[1]
4. We demonstrate that DGE's ability to train accurate and diverse classifiers is central to strong performance, and show that ensembling multiple GNNs with separate parameters is a consistent way to maximize the trade-off between accuracy and diversity.

## 2 BACKGROUND AND PRELIMINARIES

### 2.1 HIGHER-ORDER NETWORKS

Let $\mathcal{S} = \{S_1, S_2, ..., S_n\}$ be a set of **observed paths** (e.g., flight itineraries, disease trajectories, or user clickstreams), where each $S_i = \langle s_1, s_2, ..., s_m \rangle$ is a sequence of **entities** (e.g., airports, diagnosis codes, or web pages). Let $\mathcal{A} = \bigcup \mathcal{S}$ denote the set of entities across all sequences. By using a graph to summarize $\mathcal{S}$, we can model the global function of each entity in the system and solve a number of useful learning problems. For example, we can predict disease function via node classification or interactions between airports using link prediction.[2] However, there is a large space of possible graphs that can represent $\mathcal{S}$. We consider two: a FON, and the HON introduced by Xu et al. (2016). In a FON $\mathcal{G}_1 = (\mathcal{V}_1, \mathcal{E}_1)$, the node set $\mathcal{V}_1 = \mathcal{A}$ (or, more generally, the mapping $f : \mathcal{V}_1 \to \mathcal{A}$ is bijective), and the edge set $\mathcal{E}_1$ is the set of node pairs $(u, v) \in \mathcal{V}_1 \times \mathcal{V}_1$ that are adjacent elements in at least one $S_i$.

---

[1]Code and 3 data sets are available at `https://github.com/sjkrieg/dge`.

[2]This is distinct from sequence models like transformers, which typically predict an entity's local function within a single sequence.

In a HON $\mathcal{G}_k = (\mathcal{V}_k, \mathcal{E}_k)$ with order $k > 1$, each node is a sequence of entities $u' = \langle a'_1, ..., a'_{m-1}, a'_m \rangle$, where each $a'_i \in \mathcal{V}_1$ and $m \leq k$. We define $a'_m$ as the **base node**, and in practice use the notation $u' = a'_m | a'_1, ..., a'_{m-1}$ to emphasize that each $u' \in \mathcal{V}_k$ represents a base node whose current state is conditioned on a set of predecessors. Each node can have a different number of predecessors, and a conditional node with $m > 1$ is only created if the conditional distribution of paths it encodes sufficiently reduces the entropy of the graph (Saebi et al., 2020d). This means that $\mathcal{V}_k \subseteq \mathcal{A}^k$; or, more generally, the mapping $f : \mathcal{V}_k \to \mathcal{A}^k$ is injective but not necessarily bijective. We define $\Omega_u^k = \{ u' \in \mathcal{V}_k, a'_m = u \}$ as the **higher-order family** of $u$ (including $u$ itself), and call each $u' \in \Omega_u^k$ a **relative** of $u$ (in Figure 1, for example, C|A and C|B are the relatives of C). Like $\mathcal{E}_1$, the edge set $\mathcal{E}_k$ is the set of node pairs $(u, v) \in \mathcal{V}_k \times \mathcal{V}_k$ that are adjacent in at least one $S_i$. In both HONs and FONs, edges are directed such that $(u, v) \neq (v, u)$ and weighted via $w_k : \mathcal{E}_k \to \mathbb{R}_{\geq 0}$, where 0 indicates a missing edge. By creating conditional nodes, a HON can express higher-order interactions while remaining a graph, since each edge is still a 2-tuple. For example, consider that passengers who fly from Atlanta to Chicago are much more likely to fly back to Atlanta than to New York, and vice versa. A HON can encode this dependency by creating the conditional nodes "Chicago|Atlanta" and "Chicago|New York", which changes the topology and flow of information within local neighborhoods (Rosvall et al., 2014). Choosing which conditional nodes and edges to create is a non-trivial problem; readers interested in more details can refer to Krieg et al. (2020a), who proposed the procedure that we used in this study.

**Related graph-based models** The term "higher-order" is also used in the literature to refer to the analysis of polyadic structures within graphs (Benson et al., 2016), as well GNNs that are able to distinguish these structures (Morris et al., 2019; Li et al., 2020; Schnake et al., 2021). These studies rely on the same assumptions as other GNNs, i.e., that a graph is a sufficient approximation of the neighborhood structure. Despite similarities in terminology, HONs are primarily concerned with the question of initial representation (i.e., how should the graph be constructed?) rather than downstream analysis of an existing graph, and thus address a fundamentally different—though complementary— problem (Lambiotte et al., 2019). HONs are also distinct from other formalisms like hypergraphs and simplicial complexes in that they encode conditional distributions that govern higher-order paths (via conditional nodes and directed, weighted edges), and therefore represent different kinds of systems (Battiston et al., 2020; Porter, 2020; Torres et al., 2021; Battiston et al., 2021). Some very recent works have shown that aggregators based on paths (via random walks) can improve the expressiveness of GNNs (Eliasof et al., 2022; Jin et al., 2022b), but these still rely on the graph structure to guide path sampling. This further motivates the use of representations like HONs, which are designed to consistently and accurately encode higher-order paths for downstream learning tasks (Rosvall et al., 2014; Xu et al., 2016; Saebi et al., 2020d; Krieg et al., 2020a). To our knowledge, only one study has previously used GNNs with HONs (Jin et al., 2022a); however, as we discuss in Section 3, its proposed method has critical shortcomings.

## 2.2 GRAPH NEURAL NETWORKS

A generic GNN computes a hidden vector representation for a node $u$ at timestamp $t$ according to:

$$\mathbf{h}_u^{(t)} = \text{COMBINE}\left( \mathbf{h}_u^{(t-1)}, \text{AGGREGATE}\left( \{ \mathbf{h}_v^{(t-1)}, v \in \mathcal{N}(u) \} \right) \right), \qquad (1)$$

where $\mathcal{N}(u)$ is the neighborhood of $u$ in a graph $\mathcal{G}$. What typically distinguishes GNNs is how they define COMBINE and AGGREGATE, and how they represent $\mathcal{N}(u)$ (Xu et al., 2019; Zhou et al., 2020; Wu et al., 2020). By recursively pooling features via Eq. 1, GNNs implicitly construct higher-order neighborhoods across transitive paths. This allows nonadjacent nodes to share information if they are close in the graph (Li et al., 2018; Chen et al., 2020); however, as we will demonstrate, when higher-order dependencies are present, assuming transitivity leads GNNs that are trained on FONs to generalize poorly.

In this work, we do not reformulate Eq. 1 and are agnostic toward its particular implementation. We instead abstract GNN as a function that takes a single node $u$ as input and returns either the final hidden representation $\mathbf{h}_u^{(t)}$, or, in a supervised setting, a vector of predicted label probabilities $\hat{\mathbf{y}}_u$:

$$\text{GNN}(u) = \mathbf{h}_u^{(t)} \quad \text{or} \quad \text{GNN}(u) = \hat{\mathbf{y}}_u. \qquad (2)$$

We always assume that $\mathcal{G}$ contains initial node features $\{ \mathbf{x}_u, \forall u \in \mathcal{V} \}$ and that $\text{GNN}(\cdot)$ is parameterized by weights $\theta$, but for simplicity we omit them from our notation.

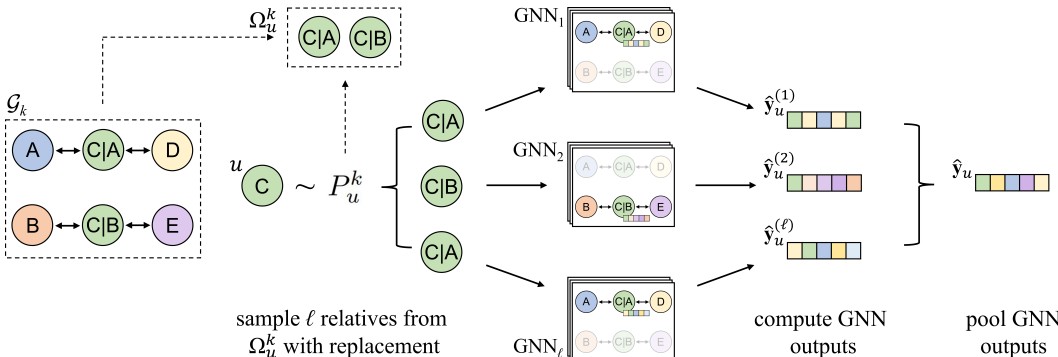

Figure 2: Overview of DGE. Given a HON $\mathcal{G}_k$, DGE computes outputs for each base node $u$ by a) resampling relatives from $u$'s higher-order family $\Omega_u^k$ via a sampling distribution $P_u^k$, b) computing outputs for each sampled relative using independent GNN modules, and c) pooling the outputs.

**GNN ensembles** Ho (1998) and Breiman (2001) showed that training an ensemble of shallow learners on random subspaces could exploit variance in the feature space and improve performance, and Dietterich (2000) demonstrated that these ensembles are most potent when the predictions of their base classifiers are accurate and diverse. Deep ensembles have typically been used with random weight initializations to improve uncertainty estimation and robustness (Lakshminarayanan et al., 2017; Fort et al., 2019; Wasay & Idreos, 2020), which has benefited GNNs via mechanisms like multi-head attention (Veličković et al., 2018; Brody et al., 2022; Hou et al., 2021), but very few works have directly explored ensembles of GNNs. Some recent exceptions have suggested that ensembling subgraphs could benefit GNNs (Zeng et al., 2021; Tang et al., 2021; Lin et al., 2022).

## 3 DEEP GRAPH ENSEMBLES FOR HIGHER-ORDER NETWORKS

### 3.1 WHY ENSEMBLES?

There are a number of design challenges (**CHs**) we must address in order to realize the joint potential of GNNs and HONs. In a HON, entities are represented by a non-fixed number of conditional nodes (**CH1**); for example, in Figure 1, C is represented by two nodes but A, B, D, and E are each only represented by one node. Conditional nodes typically have different neighborhoods (**CH2**); for example, in Figure 1, C|A and C|B have different neighbors. Further, they may vary in importance (i.e., degree) in the graph (**CH3**) (Xu et al., 2016; Saebi et al., 2020d). To address these challenges, one intuitive idea is to reformulate Eq. 2 so that it computes a representation for $u$ by sampling neighbors from any of $u$'s relatives. However, this fails to address CH2 because a GNN would aggregate the samples without considering differences between relatives. Another idea is to compute representations for each relative separately, then pool them via a permutation-invariant function like an elementwise MEAN, as proposed for HO-GNN by Jin et al. (2022a). But this does not address CH3, since all relatives would contribute equally to the final representation. Moreover, if we assume that all relatives in a higher-order family share the same (or similar) features, this solution will overrepresent features associated with larger higher-order families. In order to propose a method that comprehensively addresses these challenges, we first consider the following relationship.

**Theorem 1.** *Let $\mathcal{G}_1$ and $\mathcal{G}_k$ be a FON and HON, respectively, both constructed from the same input $\mathcal{S}$. Let $\mathcal{N}_1(u)$ and $\mathcal{N}_k(u)$ denote the neighborhoods of any node $u$ in $\mathcal{G}_1$ and $\mathcal{G}_k$, respectively. Let* AGGREGATE$(\cdot)$ *represent any symmetric neighborhood aggregation function. If $u \in \mathcal{V}_1$ and $u' \in \Omega_u^k$, then* AGGREGATE$(\mathcal{N}_k(u'))$ *is a biased estimator of* AGGREGATE$(\mathcal{N}_1(u))$.

We prove Theorem 1 in Appendix A. Intuitively, we observe that HONs are constructed such that $\mathcal{N}_k(u') \subseteq \mathcal{N}_1(u)$, and $u'$ only exists in $\mathcal{G}_k$ if the expectation of a random walker differs substantially (measured via KL-divergence) from $u$ in $\mathcal{G}_1$ (Saebi et al., 2020d; Krieg et al., 2020a). Consequently, these differences in neighborhood structure will shift the expectation of the features gathered by AGGREGATE$(\mathcal{N}_k(u'))$. A typical strategy for training a single GNN would involve attempting to eliminate this variance via some sampling method on the graph. We instead take an ensemble

approach and propose to regularize the model via multiple GNNs that err in different ways. Toward this end, and inspired by feature subspace methods (Ho, 1998; Breiman, 2001), we call $\mathcal{N}_k(u')$ a **neighborhood subspace** of $\mathcal{N}_1(u)$.

**Remark**    Our result from Theorem 1 also relates to the expressiveness of an aggregator over neighborhoods in $\mathcal{G}_k$. Consider two nodes $u, v \in \mathcal{V}_1$ whose rooted subgraphs (i.e., neighborhoods) are non-isomorphic but are not distinguishable by the aggregator in $\mathcal{G}_1$. As long as there exists some $u' \in \Omega_u^k$ (excluding $u$ itself), then we know that, since it is a biased estimator, the aggregator can distinguish $u'$ from $u$ in $\mathcal{G}_k$. It follows transitively that the aggregator can also distinguish $u'$ from $v$. It is possible that there also exists some $v' \in \Omega_v^k$ that cannot be distinguished from $u'$, but this would be extremely unlikely to occur over all pairs of relatives in $\Omega_u^k$ and $\Omega_v^k$. We can thus improve the expressiveness of the model by allowing the information from nodes in $\Omega_u^k$ to contribute to the final representation for each $u$.

Guided by these observations, we propose to address the CHs outlined above by training an ensemble of GNNs $\{\text{GNN}_1, \text{GNN}_2, ..., \text{GNN}_\ell\}$. Given a set of training nodes $D \subseteq \mathcal{V}_1$, we generate bootstraps $\{D^{(1)}, D^{(2)}, ..., D^{(\ell)}\}$ subject to the constraint that $|D^{(i)} \bigcap \Omega_u^k| = 1$ for all $u \in D$ and $i \leq \ell$ (i.e., each bootstrap contains exactly one relative of each training node). This constraint allows us to avoid the feature overrepresentation problem, address CH1 by sampling with replacement, and address CH2 by training each $\text{GNN}_i$ on different neighborhood subspaces as represented in $D^{(i)}$. To solve CH3, we weight the sampling probability for each relative $u'$ according to the normalized out-degree of its higher-order family:

$$P_u^k(u') = \frac{\text{OUTDEG}_k(u')}{\sum_{v' \in \Omega_u^k} \text{OUTDEG}_k(v')}, \tag{3}$$

where $u \in D$ and $\text{OUTDEG}_k(u')$ is the weighted out-degree of $u'$ in $\mathcal{G}_k$. Because weighted out-degree in $\mathcal{G}_k$ is the frequency with which $u'$ appears in $\mathcal{S}$ (Krieg et al., 2020a), it is a natural measure of the importance of $u'$ with respect to the rest of the higher-order family $\Omega_u^k$. If $D \subseteq \mathcal{E}_1$ consists of node pairs for an edge task, we modify Eq. 3 slightly. For each edge $(u, v) \in D$, we resample a single pair of relatives $(u', v')$ with probability according to the normalized weights of all edges between relatives of $u$ and $v$:

$$P_{u,v}^k(u', v') = \frac{w_k(u', v')}{\sum_{(u'', v'') \in \Omega_u^k \times \Omega_v^k} \left( w_k(u'', v'') \right)}. \tag{4}$$

## 3.2    TRAINING AND INFERENCE

Let $D_u^{(i)}$ denote the relative of $u$ that was sampled for the $i^{th}$ bootstrap. We consider a supervised or semisupervised setting, in which our goal is to predict class probabilities $\hat{\mathbf{y}}_u$ for each $u \in D$ such that some loss is minimized w.r.t. ground truth $\mathbf{y}_u$. We propose three methods for computing $\hat{\mathbf{y}}_u$:

$$\hat{\mathbf{y}}_u = \sigma\Big(\text{CONCAT}\big(\{\text{GNN}_i\,(D_u^{(i)}), \forall i \leq \ell\}\big)^\top \cdot \mathbf{W}\Big) \quad \text{and} \quad \text{GNN}_i(u') = \mathbf{h}_{u'}^{(i,t)}, \tag{5a}$$

$$\hat{\mathbf{y}}_u = \sigma\Big(\text{MEAN}\big(\{\text{GNN}_i(D_u^{(i)}), \forall i \leq \ell\}\big)^\top \cdot \mathbf{W}\Big) \quad \text{and} \quad \text{GNN}_i(u') = \mathbf{h}_{u'}^{(i,t)}, \tag{5b}$$

$$\hat{\mathbf{y}}_u = \text{MEAN}\Big(\{\text{GNN}_i(D_u^{(i)}), \forall i \leq \ell\}\Big) \quad \text{and} \quad \text{GNN}_i(u') = \hat{\mathbf{y}}_{u'}^{(i)}, \tag{5c}$$

where $\sigma$ is a non-linear activation, CONCAT is vector concatenation, MEAN is elementwise mean, $\mathbf{x}^\top$ is the transpose of $\mathbf{x}$, $\mathbf{h}_{u'}^{(i,t)} \in \mathbb{R}^d$ represents the hidden state of node $u'$ in the $t^{th}$ (final) layer of $\text{GNN}_i$, and $d$ is the number of hidden units (we assume $d$ is fixed for all $\text{GNN}_i$). Using $c$ to denote the number of classes, for Eq. 5a we have $\mathbf{W} \in \mathbb{R}^{d\ell \times c}$, and for Eq. 5b we have $\mathbf{W} \in \mathbb{R}^{d \times c}$. We refer to Eq. 5a as **DGE-concat**, Eq. 5b as **DGE-pool**, and Eq. 5c as **DGE-bag**.

In DGE-concat and DGE-pool each GNN outputs hidden representations, which are concatenated or pooled, respectively, before computing the logits. This means that they can be trained in end-to-end fashion as a single neural network with parallel but independent GNN modules. During a forward pass, each $\text{GNN}_i$ only computes representations for the nodes in $D^{(i)}$. In DGE-bag, on the other hand, each GNN outputs class probabilities, which means that each GNN is trained independently

Table 1: Summary of graphs used in node classification and link prediction experiments.

| Name | $|S|$ | $|\mathcal{V}_1|$ | $|\mathcal{E}_1|$ | $|\mathcal{V}_2|$ | $|\mathcal{E}_2|$ | # classes | $\mathcal{H}(\mathcal{G}_1)$ | $\mathcal{H}(\mathcal{G}_2)$ |
|------|------|------|------|------|------|------|------|------|
| Air | 17.1m | 416 | 13,735 | 7,461 | 236,806 | 10 | 0.351 | 0.288 |
| T2D | 0.8m | 908 | 314,352 | 5,462 | 367,530 | 16 | 0.099 | 0.099 |
| Wiki | 76.2k | 4,179 | 70,662 | 5,584 | 84,806 | 10 | 0.366 | 0.375 |
| Mag | 3.8m | 4,079 | 1,873,279 | 9,568 | 1,957,602 | 4 | 0.323 | 0.336 |
| Mag+ | 8.1m | 17,428 | 5,098,787 | 192,204 | 7,980,026 | 6 | 0.292 | 0.310 |
| Ship | 54.9k | 5,586 | 369,952 | 8,586 | 422,426 | 38 | 0.477 | 0.489 |

and their predictions are simply averaged to compute the final probabilities. We also designed an attention-based pooling method, but found that it did not generalize well (Appendix D).

One important question is whether all $\text{GNN}_i$ should share parameters. In other words, is an ensemble necessary, or is it sufficient to use single, more complex model (Abe et al., 2022)? We evaluated this question experimentally, and use **DGE-concat**\*, **DGE-pool**\*, and **DGE-bag**\* to denote shared-parameter variants of Eqs. 5a, 5b, and 5c, respectively. For DGE-concat* and DGE-pool*, we adjusted our training procedure so that, during backpropagation, each parameter was updated once according to its contribution to the summed loss across all $D^{(i)}$. For DGE-bag*, this meant that each $\text{GNN}_i$ was essentially pretrained on $D^{(i-1)}$. Since each $\text{GNN}_i$ shares parameters, none of these variants are true ensembles. Instead, they are single models that synthesize representations for conditional nodes via a READOUT function (Wu et al., 2020) on each higher-order family. DGE-pool* is similar to HO-GNN (Jin et al., 2022a), which uses a single GNN and computes a representation for each $u$ via the mean of all its relatives (rather than a weighted sample) in $\Omega_u^k$. We also considered one final variant, **DGE-batch**\*, which does not use a fixed set of bootstraps for training. Instead, it uses Eq. 3 to sample a new set of relatives for each batch. Then, during inference, we use the same procedure as DGE-bag: resample $\ell$ relatives for each node and compute their outputs via Eq. 5c. DGE-batch* thus drops any resemblance to an ensemble, instead addressing CH1, CH2, and CH3 entirely via batch sampling.

Figure 2 summarizes the components of DGE: resampling $\ell$ relatives for each entity via Eq. 3, computing a node representation for each sampled relative via Eq. 2, and pooling the computed representations via Eq. 5a, 5b, or 5c. In general, DGE's computational cost is linear with the cost of the base GNN and the ensemble size $\ell$, since we are essentially constructing $\ell$ copies of that GNN (or, in the case of shared parameters, repeating $\ell$ forward passes per example). For a given node $u$, the additional cost of sampling and pooling $\ell$ relatives is $O(\ell\,|\Omega_u^k|)$ and $O(\ell)$, respectively, which are trivial compared to the costs of Eq. 2 (Wu et al., 2020). Additionally, the one-time cost of constructing $\mathcal{G}_k$ increases linearly with $k$ (Krieg et al., 2020a). Since we used $\mathcal{G}_2$ in our experiments, there was marginal overhead for graph construction as compared to $\mathcal{G}_1$.

## 4 EXPERIMENTAL RESULTS AND DISCUSSION

### 4.1 EXPERIMENTAL SETUP

We used GROWHON (Krieg et al., 2020a) to construct FONs ($\mathcal{G}_1$) and HONs with $k = 2$ ($\mathcal{G}_2$) for six real-world data sets with known higher-order dependencies: flight itineraries for airline passengers in the United States (**Air**) (Rosvall et al., 2014), disease trajectories for type 2 diabetes patients in Indiana (**T2D**) (Krieg et al., 2020b), clickstreams of users playing the Wikispeedia game (**Wiki**) (West et al., 2009), readership trajectories for a large online magazine (**Mag** and a larger version, **Mag+**, for node classification only) (Wang et al., 2020), and global shipping routes (**Ship**) (Saebi et al., 2020c). Table 1 summarizes their key characteristics, including average homophily $\mathcal{H}$ (see Appendix G for details). We discuss additional details and preprocessing steps in Appendix B.

We evaluated several baselines, including GCN (Kipf & Welling, 2017), GAT (Veličković et al., 2018), GraphSAGE (Hamilton et al., 2017), GIN (Xu et al., 2019) and GATv2 (Brody et al., 2022), and SEAL (Zhang & Chen, 2018) (link prediction only). Other noteworthy baselines are Graph-SAINT (Zeng et al., 2020) which samples a different subgraph for each training iteration (Hu et al., 2020); GCNII (Chen et al., 2020), which uses residual connections to address over-smoothing; and

Table 2: Node classification results (micro F1). Bold font indicates the best result for each data set.

| Model | Air | T2D | Wiki | Mag | Mag+ | Ship |
|---|---|---|---|---|---|---|
| GCN | $0.818 \pm 0.03$ | $0.480 \pm 0.02$ | $0.643 \pm 0.01$ | $0.796 \pm 0.01$ | $0.710 \pm 0.01$ | $0.746 \pm 0.01$ |
| GCNII | $0.839 \pm 0.05$ | $0.511 \pm 0.02$ | $0.654 \pm 0.02$ | $0.801 \pm 0.01$ | $0.712 \pm 0.01$ | $0.770 \pm 0.01$ |
| GAT | $0.804 \pm 0.03$ | $0.282 \pm 0.10$ | $0.639 \pm 0.02$ | $0.487 \pm 0.06$ | $0.511 \pm 0.02$ | $0.745 \pm 0.02$ |
| GATv2 | $0.838 \pm 0.03$ | $0.292 \pm 0.07$ | $0.643 \pm 0.03$ | $0.495 \pm 0.05$ | $0.527 \pm 0.02$ | $0.747 \pm 0.01$ |
| GraphSAGE | $0.781 \pm 0.04$ | $0.654 \pm 0.04$ | $0.625 \pm 0.02$ | $0.808 \pm 0.02$ | $0.689 \pm 0.01$ | $0.796 \pm 0.01$ |
| GIN | $0.745 \pm 0.02$ | $0.673 \pm 0.04$ | $0.636 \pm 0.02$ | $0.826 \pm 0.02$ | $0.722 \pm 0.00$ | $0.801 \pm 0.01$ |
| GraphSAINT | $0.802 \pm 0.02$ | $0.600 \pm 0.07$ | $0.664 \pm 0.01$ | $0.821 \pm 0.02$ | $0.719 \pm 0.01$ | $0.803 \pm 0.01$ |
| PathGCN | $0.846 \pm 0.02$ | $0.523 \pm 0.03$ | $0.616 \pm 0.03$ | $0.735 \pm 0.02$ | $0.622 \pm 0.01$ | $0.781 \pm 0.01$ |
| HONEM | $0.805 \pm 0.04$ | $0.566 \pm 0.02$ | $0.588 \pm 0.01$ | $0.728 \pm 0.02$ | $0.620 \pm 0.01$ | $0.750 \pm 0.01$ |
| HO-GNN | $0.822 \pm 0.02$ | $0.436 \pm 0.03$ | $0.581 \pm 0.01$ | $0.785 \pm 0.01$ | $0.702 \pm 0.01$ | $0.787 \pm 0.01$ |
| DGE-concat | $0.825 \pm 0.04$ | $0.501 \pm 0.06$ | $0.615 \pm 0.02$ | $0.790 \pm 0.02$ | $0.681 \pm 0.04$ | $0.828 \pm 0.01$ |
| DGE-concat* | $0.810 \pm 0.04$ | $0.439 \pm 0.03$ | $0.577 \pm 0.02$ | $0.761 \pm 0.02$ | $0.642 \pm 0.01$ | $0.809 \pm 0.01$ |
| DGE-pool | $0.839 \pm 0.03$ | $0.735 \pm 0.03$ | $0.671 \pm 0.01$ | $0.860 \pm 0.01$ | $0.722 \pm 0.01$ | $0.808 \pm 0.01$ |
| DGE-pool* | $\mathbf{0.865} \pm 0.02$ | $0.555 \pm 0.07$ | $0.599 \pm 0.04$ | $0.775 \pm 0.01$ | $0.671 \pm 0.01$ | $0.767 \pm 0.02$ |
| DGE-bag | $0.856 \pm 0.02$ | $\mathbf{0.770} \pm 0.04$ | $\mathbf{0.681} \pm 0.00$ | $\mathbf{0.871} \pm 0.01$ | $\mathbf{0.769} \pm 0.01$ | $\mathbf{0.840} \pm 0.01$ |
| DGE-bag* | $0.766 \pm 0.04$ | $0.719 \pm 0.04$ | $0.644 \pm 0.02$ | $0.841 \pm 0.02$ | $0.739 \pm 0.01$ | $0.825 \pm 0.01$ |
| DGE-batch* | $0.764 \pm 0.03$ | $0.646 \pm 0.01$ | $0.623 \pm 0.01$ | $0.818 \pm 0.01$ | $0.742 \pm 0.00$ | $0.812 \pm 0.01$ |

\* shared parameters

Table 3: Link prediction results (AUPRC). Bold font indicates the best result for each data set.

| Model | Air | T2D | Wiki | Mag | Ship |
|---|---|---|---|---|---|
| Best baseline | $0.818 \pm 0.02$ (SEAL) | $0.818 \pm 0.00$ (HONEM) | $0.834 \pm 0.01$ (SEAL) | $0.879 \pm 0.00$ (HO-GNN) | $0.887 \pm 0.01$ (SEAL) |
| DGE-concat | $\mathbf{0.886} \pm 0.01$ | $0.815 \pm 0.01$ | $0.774 \pm 0.01$ | $0.802 \pm 0.00$ | $\mathbf{0.910} \pm 0.00$ |
| DGE-concat* | $0.856 \pm 0.01$ | $0.779 \pm 0.01$ | $0.712 \pm 0.02$ | $0.769 \pm 0.01$ | $0.904 \pm 0.00$ |
| DGE-pool | $0.851 \pm 0.02$ | $\mathbf{0.920} \pm 0.00$ | $0.838 \pm 0.03$ | $0.913 \pm 0.00$ | $0.898 \pm 0.00$ |
| DGE-pool* | $0.845 \pm 0.01$ | $0.901 \pm 0.00$ | $0.802 \pm 0.06$ | $0.898 \pm 0.00$ | $0.815 \pm 0.00$ |
| DGE-bag | $\mathbf{0.887} \pm 0.00$ | $0.907 \pm 0.00$ | $\mathbf{0.876} \pm 0.01$ | $\mathbf{0.921} \pm 0.00$ | $0.895 \pm 0.00$ |
| DGE-bag* | $0.862 \pm 0.02$ | $0.894 \pm 0.00$ | $0.856 \pm 0.00$ | $0.916 \pm 0.00$ | $0.891 \pm 0.00$ |
| DGE-batch* | $0.853 \pm 0.03$ | $0.871 \pm 0.01$ | $0.830 \pm 0.01$ | $0.887 \pm 0.00$ | $0.865 \pm 0.00$ |

\* shared parameters

PathGCN (Eliasof et al., 2022), which learns spatial operators on paths sampled via random walks. We also evaluated two baselines designed specifically for HONs: HONEM, a matrix factorization method (Saebi et al., 2020a), and HO-GNN (Jin et al., 2022a). All baselines used $\mathcal{G}_1$ as input (we also evaluated each baseline using $\mathcal{G}_2$ as input, details are in Appendix E). We manually tuned each model (details in Appendix C). For DGE, unless noted otherwise, we fixed $\ell = 16$ and used the mean-pooling variant of GraphSAGE as the base GNN, since a) it performed reasonably well as a baseline on all six data sets, and b) its sample-and-aggregate procedure intuitively complements the relative sampling and pooling used by DGE. We used Python 3.7.3 and Tensorflow 2.4.1 for all experiments, and utilized Stellargraph 1.2.1 (Data61, 2018) for the implementation of DGE.

## 4.2 EXPERIMENTAL RESULTS

**Node classification and link prediction** On the node classification task (Table 2), DGE-bag outperformed all other methods on five data sets and was second to DGE-pool* on Air. DGE-pool performed only slightly worse than DGE-bag in all cases, but DGE-concat did not generalize well and performed poorly in all cases except Ship. This suggests that our relative sampling procedure was effective in conjunction with ensembling (DGE-bag), but not as effective at regularizing a single fully-connected network against the variance in $\mathcal{G}_k$. Since concatenation depends heavily on the position of each relative in the vector representation, DGE-concat overfit on nodes that had many relatives.

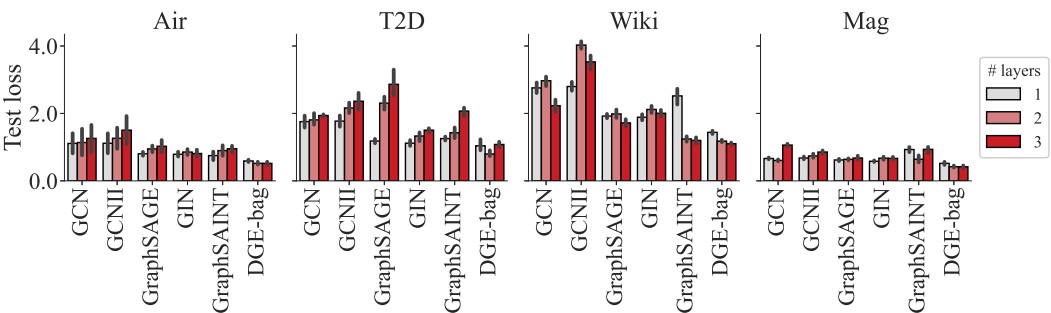

Figure 3: Node classification loss as a function of model depth. Error bars are standard deviation.

Sharing parameters typically reduced performance for each DGE variant, excepting DGE-pool* on Air. We discuss this observation in Section 4.3. The differences between HO-GNN and DGE emphasize the importance of accounting for both the variance in neighborhood subspaces in $\mathcal{G}_k$ as well as the differences in importance between relatives. Some baselines were occasionally competitive: GCNII, GATv2, and PathGCN performed relatively well on Air, and GCNII and GraphSAINT performed relatively well on Wiki. However, on Mag and especially T2D, DGE-bag outperformed all baselines by a significant margin. PathGCN and the full-batch models struggled on T2D, likely due to its high density—meaning that there are many transitive and false positive paths in $\mathcal{G}_1$. Low homophily may also have contributed to their poor performance (Zhu et al., 2020) (see discussion in Appendix G). GAT and GATv2 performed especially poorly on the dense graphs, likely because they compute attention weights (instead of using the given edge weights) and were thus more susceptible to overfitting. The link prediction results, summarized in Table 3, tell a similar story. Because of its strong performance, we focus the remainder of our analysis on DGE-bag. For details on training time and convergence for all models, please see Appendix F.

**Model size** For the baselines, increasing the number of GNN layers typically increased generalization error (Figure 3). For DGE-bag, the test loss decreased with increased model depth in all but one case (layer 3 on T2D). These results support the findings of Lambiotte et al. (2019) that transitively inferring paths in a FON cannot account for higher-order dependencies, and our hypothesis that GNNs trained on $\mathcal{G}_1$ will overfit on non-existent paths. The exception to this pattern was Wiki, on which many models performed best with 3 layers. This difference is perhaps because Wiki was the sparsest graph, meaning that the likelihood of sampling a false positive path is relatively low. However, DGE-bag still produced the lowest test error at all depths.

In order to ensure that DGE's performance was not simply due to using a more expensive model, we compared DGE-bag to the strongest baselines at several parameter budgets (Table 4; results for Wiki and Mag are available in Appendix E). DGE-bag underperformed with the smallest budgets, since each base learner was too weak and underfit. We found that sharing parameters (DGE-bag*) resolved this issue and produced strong results on T2D, Wiki, and Mag. However, with a moderate parameter budget, DGE-bag consistently outperformed other models. Again, for most baselines, increasing model depth simply caused caused them to overfit and decreased performance.

### 4.3 ENSEMBLE DIVERSITY AND PARAMETER SHARING

The variants of DGE that used separate parameters generally performed best. To understand this observation, we draw from prior work on ensembles, which has established that ensembles are most potent when the individual classifiers have low error and high disagreement (Dietterich, 2000). As Figure 4 shows, DGE-bag consistently produced classifiers that were both diverse and accurate. The shared-parameter variants often produced classifiers that were either diverse or accurate, but not both. We observed similar results for Air, Wiki, and Mag (Appendix E). These results reflect node classification performance (Table 2) and support several important conclusions. First, the high disagreement in DGE-bag demonstrates that treating higher-order relatives as neighborhood subspaces successfully introduced variance into the model. Second, the low mean error for DGE-bag demonstrates that ensembling GNNs with separate parameters effectively captured and exploited this variance. Third, sharing parameters (i.e., using a single, more complex model) did not achieve the same

Table 4: Node classification results (mean micro F1 for 5-fold cross validation) under various parameter budgets. Bold font indicates the best result for each budget and data set.

| Model | Layers | Total parameters (Air) | | | | | Total parameters (T2D) | | | | |
|---|---|---|---|---|---|---|---|---|---|---|---|
| | | 10k | 50k | 100k | 500k | 1m | 10k | 50k | 100k | 500k | 1m |
| GCNII | 2 | **0.78** | **0.79** | 0.80 | 0.80 | 0.80 | 0.43 | 0.47 | 0.47 | 0.49 | 0.50 |
| | 4 | 0.77 | 0.78 | 0.78 | 0.79 | 0.79 | 0.43 | 0.47 | 0.50 | 0.51 | 0.51 |
| | 8 | 0.74 | 0.76 | 0.76 | 0.76 | 0.76 | 0.45 | 0.49 | 0.49 | 0.51 | 0.51 |
| GATv2 | 2 | 0.66 | 0.73 | 0.80 | 0.82 | 0.83 | 0.11 | 0.14 | 0.17 | 0.18 | 0.18 |
| | 4 | 0.49 | 0.58 | 0.62 | 0.71 | 0.75 | 0.10 | 0.12 | 0.14 | 0.14 | 0.15 |
| GIN | 2 | 0.74 | 0.74 | 0.75 | 0.75 | 0.75 | 0.54 | 0.64 | 0.67 | 0.66 | 0.67 |
| | 4 | 0.63 | 0.72 | 0.73 | 0.73 | 0.73 | 0.36 | 0.38 | 0.38 | 0.44 | 0.46 |
| GraphSAINT | 2 | 0.71 | 0.75 | 0.75 | 0.76 | 0.77 | 0.45 | 0.56 | 0.59 | 0.60 | 0.61 |
| | 4 | 0.60 | 0.64 | 0.68 | 0.74 | 0.74 | 0.25 | 0.27 | 0.28 | 0.28 | 0.28 |
| DGE-bag | 2 | 0.68 | **0.79** | **0.81** | **0.83** | **0.86** | 0.47 | 0.66 | **0.73** | **0.75** | **0.77** |
| DGE-bag* | 2 | 0.70 | 0.76 | 0.80 | 0.81 | 0.81 | **0.65** | **0.70** | 0.70 | 0.71 | 0.71 |

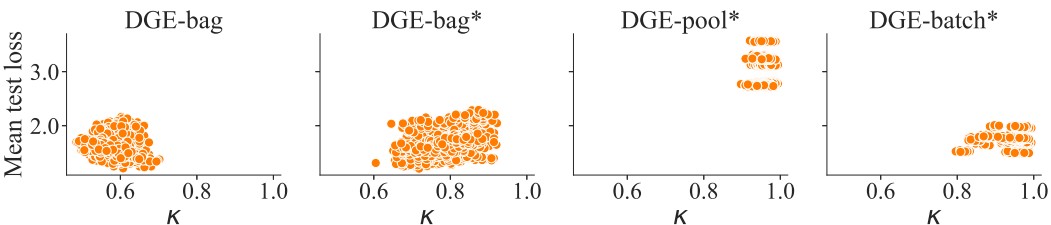

Figure 4: Mean node classification loss for all pairs of the $\ell = 16$ base GNNs within each testing fold on T2D, plotted as a function of Cohen's kappa (lower values indicate lower agreement). Each point represents one pair of GNNs in the ensemble. All plots contain the same number of points.

effect. DGE-pool* outperformed DGE-bag on Air because the classifiers were accurate enough to make up for the lack of disagreement. This may be because Air had larger higher-order families than other data sets, so each bootstrap was less representative of the true neighborhood. However, using separate parameters for each GNN—a true ensemble approach—was the most consistently effective way to balance the trade-off between accuracy and diversity.

## 5 LIMITATIONS, OTHER RELATED WORK, AND FUTURE DIRECTIONS

DGE's empirical success has implications for multiple research areas. Most broadly, we suggest that ensembling techniques for GNNs have been underexplored. However, generalizing our findings is complicated by the fact that DGE relies on the HON to supply the neighborhood variance, which is in turn limited to representing sequential data. Our results also suggest that the task of designing informed graph representations like HONs is essential and has been previously overshadowed by downstream learning algorithms. Other research on higher-order models has made progress toward this end, but much work remains to generalize these models and the types of dependencies they can represent. One area we consider promising for future work is graph structure learning. At present, HONs are constructed in unsupervised fashion, but future work could overcome this limitation by inferring the graph structure that best solves the given task (Brugere & Berger-Wolf, 2020; Chen & Wu, 2022). Unfortunately, most standard data sets for GNN tasks are pre-constructed graphs, meaning that any higher-order dependencies (sequential or otherwise) have already been lost before learning begins. By integrating graph construction into the learning process and continuing to develop more powerful GNNs, we will increase our capacity to model higher-order relationships and expand the frontier of machine learning on graphs.

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
