# OpenReview forum: "Deep Ensembles for Graphs with Higher-order Dependencies"
_ICLR.cc/2023/Conference — ICLR 2023 poster_

### Official Review · Reviewer_Ku1Z · 2022-10-23

**Confidence:** 3
**Correctness:** 4
**Technical Novelty And Significance:** 3
**Empirical Novelty And Significance:** 2
**Recommendation:** 6

**Clarity, Quality, Novelty And Reproducibility:**

This paper is easy to understand and well-organized. It's novel since few previous works discussed the correlation between high-order relationships and ensemble methods. The authors also provide the code and data they used.

**Strength And Weaknesses:**

The high-level idea is easy to understand, traditional GNN work can't handle the high-order relationship. In this work, the authors proposed a new ensemble method, which indicates that the aggregation of the target node's neighbor is a biased estimator of the aggregation of the target node. The proof is also technically sound. The paper also builds a connection between ensemble methods and GNN, which can be useful to large-scale graphs in future research.

Questions:

Actually, the high-order relationship (i.e., high-order WL test) has been discussed in many previous works. Previous work, e.g., [1], [2], and other works, both discuss the high-order relationship. These works already show better performance compared with the 1-WL model(e.g., GIN, GCN, ...). However, the problem is that we don't have enough computational resources for the higher-order relationship(since we need to sample a subgraph for a target node, which is both time and space consuming). Thus, I wonder does the sampling part in this paper share the same limitations. Or maybe it's more efficient/faster than other high-order WL test based models.

Another concern is the expression power of the proposed model. More discussion regarding the expression power will be appreciated. By far we only know that it's a biased estimator.

[1] Link Prediction Based on Graph Neural Networks
[2] Distance Encoding: Design Provably More Powerful Neural Networks for Graph Representation Learning

**Summary Of The Paper:**

This paper discusses and proposed one ensemble method of GNN. The key idea is based on the high-order relationship of a target node. The proposed idea is relatively easy to understand and implement, which indicates that future researchers can easily generalize it. The proposed methods also outperform baselines.

**Summary Of The Review:**

It's a good paper for me. The paper provides some insight and builds the connection between high-order relationships and ensemble methods. But I think more discussion is needed. Meanwhile, the experiment part only includes the 1-WL test model, which is not persuasive since it's obvious a higher-order relationship will have a better performance. Also, the runtime comparison may also show the advantage of the proposed model.

---

> ### Author Response · Authors · 2022-11-18
> **Author response**
>
> Thank you for your input and questions, especially from the perspective of model expressiveness and the relationship to models based on a higher-order WL test. Please see our responses below.
>
> > Actually, the high-order relationship (i.e., high-order WL test) has been discussed in many previous works. Previous work, e.g., [1], [2], and other works, both discuss the high-order relationship. These works already show better performance compared with the 1-WL model(e.g., GIN, GCN, ...). However, the problem is that we don't have enough computational resources for the higher-order relationship(since we need to sample a subgraph for a target node, which is both time and space consuming). Thus, I wonder does the sampling part in this paper share the same limitations. Or maybe it's more efficient/faster than other high-order WL test based models.
>
> > Another concern is the expression power of the proposed model. More discussion regarding the expression power will be appreciated. By far we only know that it's a biased estimator.
>
> Although there are some overlaps between our work and models based on the WL-test, the key difference is that we are addressing the higher-order relationships at the data (graph) level, not just at the model (learning algorithm) level. In fact, models like SEAL [1] or DE-GNN [2] would likely perform really well as base GNNs in our model. But, as you mentioned, the issue is the computational cost. In our case, the cost of constructing and using the higher-order network representation, as well as sampling and pooling relatives, is small compared to the cost of the GNN itself, which we discuss at the end of Section 3. So from that perspective, it’s much more efficient than models based on higher-order WL tests. Although it has this advantage, it is also more limited since HONs are designed for path-based data. We revised some of our comments in Section 2.1 to also reference DE-GNN [2] (which is a helpful addition) and clarify this relationship. We focused on experimental comparison with models based on the 1-WL test because of our focus on showing that HONs help address some fundamental, data-level issues that impact all GNNs. Since the GNN in our model is also based on a 1-WL test, it is not obvious that using a HON as the graph representation will improve performance - at least, this has not been established in any prior works. In fact, we hope our contributions in this work encourage future work on the synergies between GNNs based on the higher-order WL test, and graph representations like HONs. We revised some of our comments in Section 2.1 to indicate that these are different, but complementary problems.
>
> We also added a remark following Theorem 1 to discuss the expressiveness of our model, which we think bridges the gap between Theorem 1 and model expressiveness. We also think this additionally clarifies some of our motivation for design choices in Section 3, so thank you for this suggestion.

---

### Official Review · Reviewer_tGTJ · 2022-10-26

**Confidence:** 3
**Correctness:** 3
**Technical Novelty And Significance:** 2
**Empirical Novelty And Significance:** 2
**Recommendation:** 3

**Clarity, Quality, Novelty And Reproducibility:**

**************Clarity**************

The paper is clearly written to understand overall method.

**************Quality**************

More baselines have to be included to show the significance of proposed models.

**********Novelty**********

The proposed method has a limited novelty.

******************************Reproducibility******************************

The authors share their source code in the supplement. The paper has good reproducibility.

**Strength And Weaknesses:**

**Strengths**

(1) This paper addresses interesting research topic : higher-order networks (HONs), which encoder sequential higher-order dependencies.

**Weakness**

(1) I think that the novelty of this paper is limited. The proposed method is similar to path-based graph neural networks (e.g., pathGCN [1]) or random-walk based graph neural networks (e.g., RAW-GNN [2]).

(2) Also, the authors construct graph from the sequential data (paths) and then use GrowHON to construct HONs. I think HON seems very similar to paths (a sequence of nodes). Please clarify what the difference is between HON and paths.

(3) Since the input datasets used in experiments are sequential data, it would be better to compare proposed DGE with the model for representing sequential data. Also, more comparison with path-based graph neural networks such as pathGCN [1] have to be included.

(4) Can you show qualitative analysis for DGE to verify why it is appropriate approach to deal with HON?

---

[1] Eliasof, Moshe, Eldad Haber, and Eran Treister. "pathGCN: Learning General Graph Spatial Operators from Paths." ICML 2022.

[2] Jin, Di, et al. "RAW-GNN: RAndom Walk Aggregation based Graph Neural Network." IJCAI 2022.

**Summary Of The Paper:**

The paper proposed Deep Graph Ensemble (DGE) to learn effective representations of higher-order networks (HONS). HON is the networks that encode sequential higher-order dependencies. The authors conduct node classification and link prediction to evaluate DGE in comparison to graph neural networks on six datasets.

**Summary Of The Review:**

Overall, I am leaning towards rejection. My major concern is the novelty and the lack of comparison with related works. If you address my concerns, I will raise my score.

---

> ### Author Response · Authors · 2022-11-18
> **Author response**
>
> Thank you for your feedback, and in particular the connections you made between our work and some recent work on path-based GNNs, which gave us an opportunity to clarify our contributions in relation to this vein of work. Please see our response to your feedback below.
>
> > I think that the novelty of this paper is limited. The proposed method is similar to path-based graph neural networks (e.g., pathGCN [1]) or random-walk based graph neural networks (e.g., RAW-GNN [2]).
>
> Thank you for drawing our attention to these papers. Both pathGCN and RAW-GNN were published on July of this year, less than two months before we submitted the proposed work, and we were not aware of them at the time of submission. After reviewing both carefully for this revision, we posit that they actually emphasize (rather than diminish) our contributions. Both of these works emphasize that aggregating features over paths on the graph, rather than just the Laplacian matrix, can improve model expression and performance. But this means that it is even more important that paths sampled from a graph represent true interactions between nodes. Our proposed model does not do any path sampling and simply uses a generic message passing GNN. The foundational work on HONs was motivated by a desire to provide an accurate and expressive encoding of a set of paths, and this benefit is well established in several of the references we provide in Section 1. In particular, works such as Rosvall 2014 and Xu 2016 show that random walks on graphs often produce underfit paths. This would limit the performance of models like pathGCN or RAW-GNN. Therefore, we suggest that our contributions are valuable to the stream of research that is focusing on path-based GNNs. It is also likely that there would be benefits from using a path-based GNN as the base GNN in our proposed model. In our revision, we reference pathGCN and RAW-GNN in relationship to ours in Sections 2.1 and Section 5, and experimentally compare to pathGCN in Section 4.
>
> > Also, the authors construct graph from the sequential data (paths) and then use GrowHON to construct HONs. I think HON seems very similar to paths (a sequence of nodes). Please clarify what the difference is between HON and paths.
>
> HONs encode higher-order dependencies (i.e., conditional distributions that are observed in paths) in a graph. So they are similar to paths sampled from a typical graph, but they encode more (higher-order) information. For example, a random walker on a typical graph is a Markov chain. But a random walker on HONs is not limited by the Markov property and can more accurately predict/reproduce paths of length longer than 2, as shown in works such as Xu 2016 and Krieg 2020. In other words, HONs are graphs that use conditional nodes to accurately encode longer paths.
>
> > Since the input datasets used in experiments are sequential data, it would be better to compare proposed DGE with the model for representing sequential data. Also, more comparison with path-based graph neural networks such as pathGCN [1] have to be included.
>
> Most sequential models focus on problems like sequence classification, or classification of tokens within a sequence, and are not designed for global structure problems like node classification and link prediction. We comment on this in Section 2.1. Our revision is updated to include a discussion of pathGCN and RAW-GNN in Section 2.1 and Section 5. We additionally evaluated pathGCN experimentally and included the results in Table 2 and the discussion in Section 4. We were not able to experimentally evaluate RAW-GNN (the authors did not provide code, and we did not have time in this revision to implement and validate the model), but we think the results for pathGCN and other added discussion are representative of our work in relationship to other path-based models like RAW-GNN.
>
> > Can you show qualitative analysis for DGE to verify why it is appropriate approach to deal with HON?
>
> We included some examples of how DGE effectively addresses the characteristics of HONs in Section 3.1, which motivated the design of our proposed approach. Although we would like to include further qualitative analysis, we are limited by manuscript length and decided to focus on other ways to improve the paper.

---

### Official Review · Reviewer_mus1 · 2022-10-28

**Confidence:** 4
**Clarity, Quality, Novelty And Reproducibility:** N/A
**Correctness:** 2
**Technical Novelty And Significance:** 2
**Empirical Novelty And Significance:** Not applicable
**Recommendation:** 5

**Strength And Weaknesses:**

Title and abstract looked interesting



1. Just given a normal graph from arbitrary domains, what do you do?
2. The conditional node, how do you decide if you want to condition?
3. What's the benefit of condition? The similar argument doesn't make sense
4. For regular gnn, how do you use ensemble to make them better?

**Summary Of The Paper:**

This paper proposes an ensemble of GNNs that exploits variance in the neighborhood subspaces of nodes in graphs with higher-order dependencies and consistently outperforms baselines on semisupervised and supervised learning tasks.

The title and abstract looked interesting so I was expecting to see some interesting results, but the content is quite disappointing. The method/theory is too vague to be useful. Authors should answer my following concerns

1. Just given a normal graph from arbitrary domains, what do you do?
2. The conditional node, how do you decide if you want to condition?
3. What's the benefit of condition? The similar argument doesn't make sense
4. For regular gnn, how do you use ensemble to make them better?

**Summary Of The Review:**

This paper proposes an ensemble of GNNs that exploits variance in the neighborhood subspaces of nodes in graphs with higher-order dependencies and consistently outperforms baselines on semisupervised and supervised learning tasks.

The title and abstract looked interesting so I was expecting to see some interesting results, but the content is quite disappointing. The method/theory is too vague to be useful. Authors should answer my following concerns

1. Just given a normal graph from arbitrary domains, what do you do?
2. The conditional node, how do you decide if you want to condition?
3. What's the benefit of condition? The similar argument doesn't make sense
4. For regular gnn, how do you use ensemble to make them better?

---

> ### Author Response · Authors · 2022-11-18
> **Author response**
>
> Thank you for your feedback and questions. Please see our responses below, which we think can clarify our key motivation and contributions.
>
> > Just given a normal graph from arbitrary domains, what do you do?
>
> The problem we address includes the process of constructing the graph from some real-world observed data (a complex system). A complex system corresponding to phenomena such as transportation network or disease trajectories or percolation of information or supply chain has longer range dependencies among its components. As such, the first order Markovian assumption of dyadic interactions, alone, becomes limiting. This has been shown in Xu 2016 and other references in Section 1. To that end, this paper focuses on such complex systems and graph (or complex network) representation of such a system behavior must reflect the underlying higher-order dependencies. Such higher order graphs are the base graphs for our work. We certainly want to generalize this work to arbitrary graphs, and think the proposed work lays a foundation for doing so, but the contribution our paper makes focuses on these particular graphs that have higher-order dependencies. There are a couple of works that have tentatively suggested higher-order dependencies can be recovered from an arbitrary graph, but the current agreement in the literature (see references in Section 1, especially Lambiotte 2019) is that the process of initially constructing the graph is critical and should not be overlooked.
>
> > The conditional node, how do you decide if you want to condition?
>
> Deciding to create a conditional node is a non-trivial problem that is addressed by some prior works. We mention this in Section 2.1 (end of para 2). In summary, we compute the KL-Divergence of the distribution of neighbors from the candidate conditional node (for example, C|A) with respect to the base node (for example, C). We also discuss this in the proof and discussion of Theorem 1.
>
> > What's the benefit of condition? The similar argument doesn't make sense
>
> HONs use conditional nodes to encode the conditional (higher-order) distributions that are observed in a set of paths or trajectories. We discuss this in Sections 1 and 2, and have added a few more comments to Section 2.1 to try to better emphasize that the goal of the higher-order representation is to encode higher-order dependencies. Thank you for pointing out that this was not clear. Many of the works we cited in Section 1 highlight other benefits/advantages, but we only discussed them briefly in the Introduction. Our contributions focus on how the conditional nodes can benefit GNN performance.
>
> > For regular gnn, how do you use ensemble to make them better?
>
> We plan to generalize our findings to generic GNNs in future work, but our paper specifically focuses on graphs with higher-order dependencies, and how their characteristics can be exploited by ensembles to improve GNN performance.

---

### Official Review · Reviewer_3FqL · 2022-11-02

**Confidence:** 3
**Correctness:** 3
**Technical Novelty And Significance:** 2
**Empirical Novelty And Significance:** 2
**Recommendation:** 5

**Clarity, Quality, Novelty And Reproducibility:**

Reproducibility - Code provided by authors

Novelty and quality - Please see weakness 1

Clarity - Please see strength 2



**Strength And Weaknesses:**

**Strengths:**
1. The idea towards using GNNs to solve problem on HONs is interesting as well as identifying different ways to use the HON to construct the inputs for the GNNs
2. Most sections of the paper are well written and clear

**Weaknesses, corresponding suggestions and questions**
1. Unfortunately, to the reader it appears like the work is present as the first work of using GNN's for higher order networks and has altogether missed relevant GNNs works on Higher order networks - both comparing against them  e.g. [1]. Please compare and contrast with this work - and other appropriate related works.
2. The paragraph on total model size comparison is unclear and appears to be unfair as well. Firstly, why limit the number of GNN layers in the baseline GNN models? The provided image in Fig 4 is also hard to read - and appears like until the budget is increased considerably the baseline models appear to do better.  Moreover, it is unclear how to select the right number of GNNs ($l$) to be used in the ensemble - can this be written as a function of the graph statistics?.
3. Standard GNNs are known to suffer in the link prediction task and are not representative baselines - as has been widely studied and documented. Please add comparison with a method such as [2]
4. What happens if you perform simple GNN ensembling - same input graph for all GNNs in the ensemble - but with neighborhood sampling (using MEAN, POOL, etc) without using your proposed training procedure? How does this compare?

**Minor:**
1. A drawback of the proposed approach - from the results in the appendix - it appears like that the DGE takes longer to converge - in terms of epochs how big is the difference? Are these all with the same parameter budgets as well?
2. DGE-pool and DGE-concat (Even without shared parameters) seems to perform worse than the baselines models in most cases - is there any reasoning for this? Please address this in Section 4.3

**References:**
1. Jin, Di, et al. "Graph Neural Network for Higher-Order Dependency Networks." Proceedings of the ACM Web Conference 2022. 2022.
2. Zhang, Muhan, and Yixin Chen. "Link prediction based on graph neural networks." Advances in neural information processing systems 31 (2018).

**Summary Of The Paper:**

In this work, the authors propose Deep Graph Ensemble (DGE) to tackle the additional information available in higher order networks $-$ something which standard Message Passing GNNs methods are unable to process without loss of information. The authors propose to do this by leveraging an ensemble of standard MPNNs - which are used to solve the 3 design challenges associated with HONs  $-$ variable number of conditional nodes, different and potentially non-overlapping neighborhoods, and the importance of the nodes. The authors then present results of the DGE (with different aggregation strategies

**Summary Of The Review:**

Unfortunately, in the current state , the weaknesses of the paper out weigh the strengths. However, I am happy to increase my scores if the authors can answer the questions appropriately.

---

> ### Author Response · Authors · 2022-11-18
> **Author response**
>
> Thanks for your feedback - it was specific and helpful. Please see our responses to your specific questions/concerns below.
>
> > Unfortunately, to the reader it appears like the work is present as the first work of using GNN's for higher order networks and has altogether missed relevant GNNs works on Higher order networks - both comparing against them e.g. [1]. Please compare and contrast with this work - and other appropriate related works.
>
> We were not aware of ref [1] (HO-GNN) prior to our submission to ICLR (given its publication just prior in the Web Conference 2022); however, we did come across it after our submission, and have since reviewed the paper and run additional experiments that we have now included in our revision:  discussion and empirical comparison in Sections 2.1, 3.2, and 4.1. We have accordingly edited our text as well. We found that HO-GNN has some design limitations which are reflected in its experimental performance, and after careful evaluation we do not think it diminishes the contributions of our work. We also included comparisons to some path-based GNNs, per the suggestions of another reviewer, and are not aware of any other relevant work that we should include.
>
> > The paragraph on total model size comparison is unclear and appears to be unfair as well. Firstly, why limit the number of GNN layers in the baseline GNN models? The provided image in Fig 4 is also hard to read - and appears like until the budget is increased considerably the baseline models appear to do better. Moreover, it is unclear how to select the right number of GNNs to be used in the ensemble - can this be written as a function of the graph statistics?
>
> We have replaced Fig 4 with Table 4, which has more readable/detailed results, including additional model layers. We also added Figure 5 to the appendix to discuss the ensemble size, and removed the references to different ensemble sizes in the discussion of Table 4, since they were not particularly relevant to the result. Thank you in particular for this particularly helpful suggestion.
>
> > Standard GNNs are known to suffer in the link prediction task and are not representative baselines - as has been widely studied and documented. Please add comparison with a method such as [2].
>
> The revision includes results for SEAL on the link prediction experiments. You were correct; SEAL outperformed most other baselines on 3 of the 5 data sets.
>
> > What happens if you perform simple GNN ensembling - same input graph for all GNNs in the ensemble - but with neighborhood sampling (using MEAN, POOL, etc) without using your proposed training procedure? How does this compare?
>
> We have added this result to Appendix E (Table 7). In summary, it slightly improved the performance of the base GNN, but did not perform as well as DGE. We would have liked to include this as a point in the main manuscript, but were limited on space.
>
> > A drawback of the proposed approach - from the results in the appendix - it appears like that the DGE takes longer to converge - in terms of epochs how big is the difference? Are these all with the same parameter budgets as well?
>
> Correct; we’ve included a new comment that DGE takes longer to converge. This is mostly due to the overhead of neighbor sampling, but still is a drawback. We added some brief discussion to Appendix F about the number of epochs. In summary, DGE generally requires fewer epochs to converge, because it sees several subgraphs for each node in the same epoch and because G_k is sparser than G_1. In general, all models took longer to converge with smaller parameter budgets, but we do not have any detailed results to present about that observation.
>
> > DGE-pool and DGE-concat (Even without shared parameters) seems to perform worse than the baselines models in most cases - is there any reasoning for this? Please address this in Section 4.3
>
> DGE-pool did outperform baselines on most data sets, just not by a large margin. DGE-concat performed more poorly than expected, though. We had briefly commented on this in Section 4.2, para 1, but added a bit more detail in this revision per your suggestion.

---

### Author Response · Authors · 2022-11-18
**Author response**

Dear Reviewers,

Thank you for your time and effort in reviewing our submission and providing feedback. Your comments have helped us improve the paper in some key areas, and we think we have addressed almost all of your concerns/questions. Below is a summary of the most significant changes we’ve made to the manuscript, and we will reply below with responses to individual reviewers. Thank you again, and we look forward to the discussion period.
- Added experiments from additional baselines (HO-GNN, PathGCN, and SEAL) and discussion of other models (PathGCN, RAW-GNN, and HO-GNN) to Section 2.1 and 4.1
- Added a remark about model expressiveness in Section 3.1, after Theorem 1
- Replaced Fig 4. with Table 4 for a more thorough comparison of model sizes, and edited the “Model size” section (para 2 of Section 4.2) accordingly
- Added additional results and discussion (mostly to Appendices E and F) with respect to ensemble size, convergence time (# of epochs)
- Minor comments throughout to clarify that a significant part of our contribution is in uniquely addressing some of the limitations of GNNs at the data (graph representation) level, rather than just at the model level

---

### Comment · Area_Chair_HUGB · 2022-12-07
**Response to Author Feedback**

Dear Reviewers, thank you so much again for your time on this paper. The discussion phase is still ongoing, how does the author response and other reviews change your view of the paper?

---

> ### Author Response · Authors · 2022-12-12
> **Response to Author Feedback**
>
> Dear Reviewers and Area Chair,
>
> We incorporated the suggested revisions, and are hoping those to be considered. We hope to hear back, and engage in a dialogue.
>
> Thank you,
> Authors

---

### Decision · Program_Chairs · 2023-01-20

**Decision:**

Accept: poster

**Justification For Why Not Higher Score:**

The contribution of the work, while useful, is not ground-breaking.

**Justification For Why Not Lower Score:**

The work has all the hallmarks of a useful contribution to the community, accepting it would improve ICLR.

**Metareview: Summary, Strengths And Weaknesses:**

The reviewers were split about this paper and did not come to a consensus: on one hand they appreciated the clarity of the paper, on the other they had doubts about the baseline comparisons. After going through the paper and the discussion I have decided to vote to accept for the following reason: the main arguments against accepting made by the reviewers are 1) missing related work, 2) unfair total model size comparison, 3) missing ensemble comparison, 4) missing link prediction baseline, and 5) longer convergence. However, the authors responded to each one of these points convincingly. For 1), the reviewers pointed out HO-GNN, pathGCN, and RAW-GNN that should have been discussed, compared against, and hurts novelty. The authors pointed out the recent publication of these methods (all 2022, pathGCN and RAW-GNN were published in July, two months before the ICLR deadline). This addresses novelty. Further they have added a description of these and comparison against HO-GNN and pathGCN. This point seems resolved. For 2), the reviewers wondered if baselines are allowed to use more layers in order to match the ensemble size of the proposed approach, would they perform better? The authors show results for increased number of layers, which seem to make no difference or deteriorate the result. It seems that it is easy to overfit these datasets. For 3), the reviewers wonder if ensembling using neighborhood sampling would perform as well as the proposed method. The authors show in Appendix E that this is not the case. This result should really go in the main text somewhere, it’s a criticial point to show that the proposed method is doing more than just variance reduction via bagging. For 4), the reviewers pointed out that standard GNNs do poorly at link prediction and that a method such as SEAL should be compared against instead. The authors do this, and the method underperforms their approach. For 5), the authors agree that a downside of the approach is slower convergence. I’m glad the authors include a discussion on this. I would really like to see this mentioned in Section 5. For these reasons, the authors have convinced me the paper is worth accepting. Authors: you've already indicated that you've updated the submission to respond to reviewer changes, if you could double check their comments for any recommendation you may have missed on accident that would be great! The paper will make a great contribution to the conference!

**Note From Pc:**

if the above contains the word "oral" or "spotlight" please see: "oral" presentation means -> notable-top-5% and "spotlight" means -> notable-top-25%. As stated in our emails, we are disassociating presentation type from AC recommendations